

# Appraisal of healthcare students and professionals' knowledge and intention to educate the public regarding monkeypox (Mpox)

Malik Suliman Mohamed[1], Mona Timan Idriss[2], Nasser Hadal Alotaibi[3], Yusra Habib Khan[3] and Tauqeer Hussain Mallhi[3]

[1] Department of Pharmaceutics, College of Pharmacy, Jouf University, Sakaka, Saudi Arabia
[2] Department of Medical Sciences and Preparation Year, Northern College of Nursing, Arar, Saudi Arabia
[3] Department of Clinical Pharmacy, College of Pharmacy, Jouf University, Sakaka, Saudi Arabia

Corresponding author
Malik Suliman Mohamed,
msmustafa@ju.edu.sa

## ABSTRACT

**Background**. In light of the increasing prevalence of monkeypox (Mpox) infections worldwide, it is critical to assess the level of awareness regarding the disease and its transmission among healthcare students and professionals. Understanding the knowledge gaps within these groups is essential, as they play a vital role in infection prevention and public education.

**Objectives**. This study aimed to assess healthcare students and professionals' knowledge and intention to educate the public regarding Mpox.

**Methods**. An anonymous survey consisting of 37 items was constructed utilizing scientific information pertaining to Mpox, obtained from sources such as the World Health Organization (WHO), Centers for Disease Control and Prevention (CDC), and previous studies. The study sample comprised healthcare professionals and students who were residents of the Al-Jouf province in Saudi Arabia. Significant associations between sociodemographic data and the knowledge and intention to educate the public about Mpox were compared using statistical methods. The comparison of means for two or more than two groups were performed using Student $t$-test and one-way ANOVA, respectively. Furthermore, ordinal logistic regression was employed for further analysis.

**Results**. Out of 545 responses, 437 (80.2%) were incorporated into the final analysis. Almost half of the participants in the study were female (51.5%) and 44.2% fell within the age range of 20 to 25 years. Moreover, 49.7% of the participants were unmarried and 29.5% held a bachelor's degree. Over seven in ten participants were cognizant of the fact that Mpox is a public health emergency of international concern, whereas less than half of the participants were cognizant of the fact that Mpox has been reported in Saudi Arabia. The overall mean knowledge score was 10.40 ± 4.88 (score range: 0 to 21). Approximately 3.9% of the participants possess excellent knowledge (score > 17) and 44.9% possess average knowledge (score 12–17). The male gender, older participants, married, healthcare professionals and those in the medicine field were more knowledgeable. More than half of the study participants are willing to educate the public regarding Mpox, where overall intention score was 21.39 ± 6.84 (range 6 to 30). The female, married and those in the physiotherapy field showed more positive attitude and intention to educate the public regarding Mpox.

**Conclusions**. Despite the fact that the majority of participants had a high or moderate intention to educate the public about Mpox, their knowledge ranged from average to poor. In light of these findings, educational programs aimed at enhancing the knowledge and awareness of healthcare professionals and students regarding Mpox are required.

## INTRODUCTION

Monkeypox (Mpox) is a viral disease of animals (zoonotic disease), however, it was transmitted to humans by various ways of human/animal contact in Africa, USA and other regions (*Moore, Rathish & Zahra, 2023*). The Mpox virus is one of the four species of orthopoxvirus genus, in addition to cowpox virus, variola minor virus and variola major virus (*Petersen et al., 2019*). Although Mpox is very similar to smallpox in terms of virus structure, transmission and clinical presentation, it has less case-fatality rate than smallpox (*Chen et al., 2005*). Mpox possesses a broad host range and is known to infect a wide variety of animals with a possibility of host-switches with other poxvirus. However, it was first discovered and transmitted from monkeys to humans, hence it was named monkeypox (*Haller et al., 2014*; *Kaler et al., 2022*).

The Mpox virus was initially isolated in 1958 from Java macaques displaying symptoms of a rash and fever in captivity. It later transmitted from animals to humans, though its natural reservoir remains unidentified (*Magnus et al., 1959*; *Ophinni et al., 2022*). While the disease was first reported in humans in Africa in 1970 (*Breman et al., 1980*), it was long considered a neglected tropical disease. However, in recent years, Mpox has been detected in a wider geographic range, including non-endemic countries, leading to a global outbreak in 2022. In response to the rapid spread and escalating outbreaks, the World Health Organization (WHO) declared the Mpox outbreak a Public Health Emergency of International Concern to facilitate coordinated international efforts to control the disease (*World Health Organization, 2024b*).

Considering the fact that healthcare professionals play a key role in the diagnosis and management of Mpox, it is of major importance to be sufficiently knowledgeable about the disease attributes. An essential step for the effective infection prevention and control is to know the mode of transmission of Mpox, which is mainly *via* the close contact with infected animals/humans, often skin-to-skin contact, mucosal lesions, genital lesions, blood, saliva, upper respiratory secretions, aspiration of droplets or aerosol, animal biting, as well as hugging, massage, kissing and the sexual contact (*Alakunle et al., 2020*; *Peiró-Mestres et al., 2022*). The main and shared mean of transmission between all known methods is the close and direct contact with a lesion of virus, blood or body fluids from infected humans or animals (*Kaler et al., 2022*; *Khamees et al., 2023*). There is a doubt about the possibility of a direct viral transmission from semen, saliva, feces and other body fluids which the DNA of Mpox was detected, it may need further investigation (*Peiró-Mestres*

*et al., 2022*). Moreover, the Mpox virus has transplacental transmission, during and after birth or through a blood transfusion (*US Centers for Disease Control and Prevention, 2025*). Given the ongoing Mpox outbreak and its designation as a Public Health Emergency of International Concern by the World Health Organization (WHO) (*World Health Organization, 2024b*), healthcare students and professionals should prioritize educating the public about confirmed and potential transmission sources. This heightened focus is crucial for coordinating international responses and ensuring the community's well-being.

Insufficient knowledge about the closely related viruses can contribute to diagnostic errors. The clinical features of Mpox are almost similar to other pox-like illnesses such as smallpox and chickenpox, though they are different diseases. The healthcare professionals are badly needed to distinguish between these related diseases. The illness can range from less severe to more serious symptoms that may require hospitalization. The important and common attributes between these three pox viruses are the skin eruptions which may start with, or be followed by, fever and headache (*Di Giulio & Eckburg, 2004*). The rash can affect the whole body, notably the face, hands, feet, groin, and genital regions. Some people develop secondary bacterial infections, lymphadenopathy, proctitis, swollen glands, encephalitis, and myocarditis (*US Centers for Disease Control and Prevention, 2025*; *World Health Organization, 2024a*). The healthcare professionals often play an important role in educating the publics about the similarity between Mpox, seasonal flu and the common cold symptoms.

It is important to note that selecting the appropriate diagnostic testing for some viral diseases is challenging for healthcare professionals. Since the previous outbreak in 2003 and 2022, the clinical diagnoses of Mpox infection is facing difficulties due to its shared clinical presentations with other closely related viruses, notably the chickenpox, cowpox, disseminated zoster, generalized vaccinia, and the disseminated herpes simplex (*Babkin, Babkina & Tikunova, 2022*; *Jezek et al., 1988*; *Nalca et al., 2005*). The WHO recommends using the polymerase chain reaction for laboratory diagnoses of Mpox after clinical exclusion of other related viral and non-viral diseases such syphilis, scabies, measles, medication-associated allergies and bacterial skin infections that could be confused with the diagnoses of Mpox (*World Health Organization, 2024a*).

The availability of an effective antiviral agent is a significant challenge facing healthcare professionals. Currently there is no approved drug for the management of Mpox, as many other viral infections, it is mainly treated using supportive and symptomatic therapy (*World Health Organization, 2018*). The Mpox vaccine is recommended for those become in contact with infected individuals "within 4 days", or "within 2 weeks" if there are no symptoms. It is pertinent to mention that a previously known antiviral drug tecovirimat that has been used against smallpox could be a potential therapeutic modality for Mpox (*Kaler et al., 2022*; *World Health Organization, 2024a*).

Inadequate knowledge about human Mpox among general practitioners was identified in a recent survey analysis, highlighting a critical knowledge gap. This issue, observed in Indonesia, led to recommendations in a previous study to enhance awareness and understanding of Mpox among healthcare professionals (*Harapan et al., 2020a*). Healthcare providers, particularly those responsible for the initial management of emerging cases,

were surveyed about Mpox in Italy. The results revealed significant deficiencies in their understanding of the disease, with gaps spanning various critical areas. These findings, combined with inconsistent risk perceptions, highlight the urgent need for targeted educational initiatives to better prepare frontline medical professionals for effective Mpox management (*Riccò et al., 2022*). Similarly, research across 17 Arab countries assessed the knowledge of Mpox among healthcare personnel, including medical students and doctors. The study revealed significant gaps in understanding, particularly regarding the natural host, incubation period, and clinical signs of Mpox. Strengthening the knowledge of healthcare professionals in the region is critical for improving early detection and controlling the spread of the disease (*Swed et al., 2023*).

The current Mpox protection strategy underscores the necessity of coordinated action across multiple levels of healthcare and public health. Given the increasing burden of Mpox and the World Health Organization's designation of the disease as a Public Health Emergency of International Concern (PHEIC), the preparedness of healthcare professionals is crucial for effective and timely response. Prior research on healthcare workers' awareness and preparedness during the COVID-19 pandemic has demonstrated that knowledgeable and agile healthcare teams are essential for infection control and management (*Alrajhi et al., 2022*). However, it remains unclear to what extent healthcare professionals in the Al-Jouf region of Saudi Arabia possess adequate knowledge about Mpox infection and are prepared to support WHO's call for community protection through public education and awareness. This study aims to evaluate the existing knowledge and preparedness of healthcare professionals and students in Al-Jouf, along with their willingness to educate the public, positing that awareness gaps may be present and that focused educational initiatives could improve community defense against Mpox.

## MATERIALS AND METHODS

### Ethical considerations

The Local Committee of Bioethics (LCBE) of Jouf University reviewed and granted an ethical approval with the number: 3-04-44. Informed online consents were required from all study participants by reading the front page of the google form with the right to continue the survey or decline it in any part or any time.

### Study design and participants

A web-based cross-sectional survey (December 2022 to February 2023) was conducted among health professionals and students. The google form was distributed through social media related to health professionals of Al-Jouf region, KSA. The study instrument had brief information of research and the informed consent was obtained from all study participants.

### Survey instrument construction and validation

The study tool was developed based on the previous studies (*Fan et al., 2021*; *Mallhi et al., 2018*). The scientific information regarding Mpox was derived from the general information available on the websites of the WHO and the Centers for Disease Control and Prevention

(CDC) (*US Centers for Disease Control and Prevention, 2025*; *World Health Organization, 2024a*). The questionnaire comprised of 37 questions grouped in four sections. Section I contained socio-demographic information (six items); Section II included general awareness questions about Mpox (two items), knowledge questions (23 items), and attitude questions (six items). The questions of awareness and knowledge sections were designed in closed-ended questions (Yes/No/ Not sure, Multiple answer choices (with option "other" to add their own), while the intention section was in 5-Likert scale. Participants' intentions to educate the public about Mpox were assessed using the Theory of Planned Behavior (TPB), a well-established theoretical framework in social psychology. The TPB is a widely recognized and validated theoretical framework that has been shown to effectively predict intentions and health-related behaviors (*Ajzen, 1987*).

Knowledge-related questions were scored dichotomously, with correct answers receiving 1 point and incorrect or unsure answers receiving 0 point. The total knowledge score (range: 0–23) was categorized into 'excellent' (75% or higher), 'average' (74–50%), and 'poor' (less than 50%). Intention-related questions used a 5-point Likert scale (strongly agree to strongly disagree). Scores were inverted for negatively worded items. The total intention score (6–30) was categorized into 'high' (24+), 'moderate' (16–23), and 'low' (0–15).

The study tool was developed under opinion of researchers and subjected to content validation by independent health experts with prior experience in survey analysis. The reliability of the questionnaire was assessed by analyzing the responses of 36 participants (Coefficient alpha = 0.87). These responses were not used in the final analysis. The study tool was initially developed in English and then translated into Arabic language (forward and backward translation) by independent co-authors and finally rolled out in both English and Arabic in the same google form.

## Sampling, sample size estimation and data collection

A convenient sampling technique was used to achieve the minimal calculated sample size by adopting an approach of sending the google form survey to several social media tools of students and registered professionals working in community or public settings in Al-Jouf region of Saudi Arabia. An online calculator tool (https://www.calculator.net/sample-size-calculator.html) used the equation "$n = Z^2 \times P \times Q/e^2$" (*Pourhoseingholi, Vahedi & Rahimzadeh, 2013*) to estimate a size of 377 participants. The variables in the equation are: "e" is the desired level of precision (margin of error 5% or 0.05), "$P$" is the population proportion expected to respond 50%, "Q" is $1 - P$, "$Z$" is the $Z$ value value corresponding to the desired confidence level (1.96 for 95% confidence), and the total population is 20,000. The survey was closed after reaching the desired number of participants and the data was downloaded in Microsoft excel, subsequently cleaned based on exclusion/inclusion criteria and finally transferred to the statistical package of social sciences (SPSS version 25; SPSS, IBM Corp., Armonk, NY, USA).

## Data analysis

The descriptive and inferential statistics were performed using SPSS version 25. The participants' responses towards the closed-ended questions, multiple answer choices or 5-Likert scale were described in frequencies (N) along with percentages (%). The participants'
responses towards the knowledge and intention questions were calculated with mean (M) and standard deviation (SD). The one-way analysis of variance (ANOVA) with Tukey's *post-hoc* analysis or Student's *t*-test were employed for testing any significant associations between the continuous independent variables (socio-demographics) and dependent variables (knowledge, awareness and intention). Ordinal logistic regression (OLR) was also used to investigate the determinants of the two outcomes, knowledge and intention, each categorized into three groups: "excellent knowledge, average knowledge, and poor knowledge" and high intention, moderate intention, and low intention" respectively. The assumptions of the OLR were investigated before carrying out the analysis. The independent variables included in the OLR model were: age, gender, marital status, current role, education level, and field of education. These variables were included because it was assumed that the demographic variables of the survey could directly or indirectly influence the participants' knowledge and intention. The $p$-values $\leq 0.05$ were considered statistically significant throughout the analysis.

## RESULTS

Of the 545 total responses collected, 437 were included in the final analysis (response rate: 80.2%). The initial 36 responses were excluded as they were used to validate the questionnaire, while 72 responses were excluded because the respondents were unemployed. Almost half of the participants 225 (51.5%) were females, 193 (44.2%) aged between 20 –25 years old, 217 (49.7%) were single and 129 (29.5%) had bachelor degree (Table 1).

### Participants' awareness and knowledge about Mpox

Less than half of the participants ($n = 197$, 45.1%) were aware that the Mpox has been reported in Saudi Arabia, while more than seven in 10 participants 337 (77.1%) were aware that Mpox is a public health emergency of international concern. Likely, more than seven in 10 participants 324 (74.1%) recognized that Mpox is a viral infection. The participants' correct answers across all the six knowledge questions regarding the transmission of Mpox ranged between 61.6% and 31.1%. Among the ten knowledge questions regarding the signs and symptoms of Mpox, the most correctly answered question was "rash is a symptom of Mpox" 64.8%, and the least answered question was "nausea is a symptom of Mpox" 18.3%. In addition, the majority of the participants ($n = 352$, 80.5%) knew that there are recommended prevention steps to protect people from Mpox. Alarmingly, only 173 (39.6%) and 105 (24.0%) knew that there is no specific drug for Mpox or a vaccine for Mpox is available, respectively. Table 2 summarizes the participant's knowledge and awareness regarding the Mpox.

### Participants' intention to educate the public about Mpox

More than half of the participants (56.8%, $n = 248$) are willing to educate the public regarding Mpox. There around 6 in 10 respondents had a past experience in educating the public regarding the last pandemic COVID-19 which is a predictor for their intention. The perceived subjective norm and the perceived self-efficacy were found to had an impact in around half of participants, while the perceived behavioral control was found to had an

**Table 1 Participants' socio-demographics (N = 437).**

| Variables | Frequency (N) | Percentage (%) |
|---|---|---|
| Gender | | |
| Male | 212 | 48.5% |
| Female | 225 | 51.5% |
| Age in years | | |
| 20–25 Years | 193 | 44.2% |
| 26–30 Years | 81 | 18.5% |
| 31–40 Years | 100 | 22.9% |
| 41–50 Years | 48 | 11.0% |
| More than 50 | 15 | 3.4% |
| Marital status | | |
| Single | 217 | 49.7% |
| Married | 186 | 42.6% |
| Divorced | 24 | 5.5% |
| Widowed | 10 | 2.3% |
| Current role | | |
| Student | 216 | 49.4% |
| Healthcare professional | 221 | 50.6% |
| Education level | | |
| Student | 216 | 49.4% |
| Bachelor | 129 | 29.5% |
| Master | 61 | 14.0% |
| Doctorate (PhD) | 31 | 7.1% |
| Field of education | | |
| Medicine | 146 | 33.4% |
| Pharmacy | 124 | 28.4% |
| Nursing | 92 | 21.1% |
| Medical Laboratory Science | 29 | 6.6% |
| Dentistry | 30 | 6.9% |
| Physiotherapy | 16 | 3.7% |

impact in more than half of participants. In addition, around two third of participants 300 (68.6%) had a positive attitude and agreed that the provision of education to the public about Mpox will help in disease prevention and control measures. Table 3 summarizes the participant's intention to educate the public about Mpox.

## Mean knowledge and intention scores among socio-demographics

The average knowledge score about Mpox across all participants was 10.40 ± 4.88 (range 0 to 21). Only 3.9% of participants demonstrated excellent knowledge (score > 17, out of 23), while only 44.9% had an average knowledge level (scoring 12–17 out of 23), and almost half of the participant (51.2%) had poor knowledge (score < 12, out of 23). When comparing knowledge levels among health professionals and students, notable variations in scores were observed across different educational backgrounds. Students achieved an average score of 9.44 ± 4.80, whereas bachelor's degree holders scored 10.64 ± 4.58, master's degree holders

**Table 2** Participants' awareness and knowledge about Mpox (N = 437).

| Respondent's awareness | Correct N (%) | Incorrect N (%) | Not sure N (%) |
|---|---|---|---|
| Mpox has been reported in Saudi Arabia✓ | 197 (45.1%) | 66 (15.1%) | 174 (39.8%) |
| Mpox is a public health emergency of international concern✓ | 337 (77.1%) | 46 (10.5%) | 54 (12.4%) |

| Respondent's knowledge | Correct N (%) | Incorrect N (%) | Not sure N (%) |
|---|---|---|---|
| Mpox is a………….. (Bacterial Infection /**Viral Infection**) | 324 (74.1%) | 63 (14.4%) | 50 (11.4%) |
| Mpox infection transfer from direct human to human contact✓ | 269 (61.6%) | 64 (14.6%) | 104 (23.8%) |
| Mpox infection transfer from animal to human✓ | 249 (57.0%) | 53 (12.1%) | 135 (30.9%) |
| Mpox infection transfer by airborneX | 136 (31.1%) | 129 (29.5%) | 172 (39.4%) |
| Mpox infection transfer sexually✓ | 186 (42.6%) | 92 (21.1%) | 159 (36.4%) |
| Mpox virus can spread through a blood transfusion✓ | 184 (42.1%) | 95 (21.7%) | 158 (36.2%) |
| Mpox can be transmitted transplacentally✓ | 147 (33.6%) | 83 (19.0%) | 207 (47.4%) |
| Fever is a symptom of Mpox✓ | 279 (63.8%) | 48 (11.0%) | 110 (25.2%) |
| Rash is a symptom of Mpox✓ | 283 (64.8%) | 41 (9.4%) | 113 (25.9%) |
| Chills is a symptom of Mpox✓ | 202 (46.2%) | 71 (16.2%) | 164 (37.6%) |
| Exhausion is a symptom of Mpox✓ | 229 (52.4%) | 51 (11.7%) | 157 (35.9%) |
| Muscle aches and backache are symptoms of Mpox✓ | 228 (52.2%) | 52 (11.9%) | 157 (35.9%) |
| Headache is a symptom of Mpox✓ | 240 (54.9%) | 53 (12.1%) | 144 (33.0%) |
| Mpox can cause Respiratory symptoms such as flu✓ | 203 (46.5%) | 75 (17.2%) | 159 (36.4%) |
| Mpox can cause Urinary symptomsX | 106 (24.3%) | 113 (25.9%) | 218 (49.9%) |
| Swollen lymph nodes are signs of Mpox✓ | 192 (43.9%) | 64 (14.6%) | 181 (41.4%) |
| Nausea is a symptom of MpoxX | 80 (18.3%) | 170 (38.9%) | 187 (42.8%) |
| Mpox symptoms usually start within (Immediately/ Few days /**3 weeks**) of exposure to the virus | 80 (18.3%) | 221 (50.6%) | 136 (31.1%) |
| Symptoms of Mpox usually last (2–5 days/**2–4 weeks**/ Very long time) | 170 (38.9%) | 119 (27.2%) | 148 (33.9%) |
| There are recommended prevention steps to protect people from Mpox✓ | 352 (80.5%) | 31 (7.1%) | 54 (12.4%) |
| Past exposure to chickenpox can provide protection against MpoxX | 126 (28.8%) | 125 (28.6%) | 186 (42.6%) |
| Specific drug for Mpox is available X | 173 (39.6%) | 85 (19.5%) | 179 (41.0%) |
| Specific vaccine for Mpox is available ✓ | 105 (24.0%) | 151 (34.6%) | 181 (41.4%) |

**Notes.**
✓, correct answer; ×, incorrect answer.
Bolded options indicate correct answers.

scored 10.69 ± 5.07, and doctorate (PhD) holders scored 13.55 ± 4.56. The differences in knowledge scores across these educational groups were statistically significant, with healthcare professionals scoring notably higher than students (*p*-value < 0.001). Moreover, significant differences were noted in knowledge levels across various demographics, including gender, age, marital status, and field of education. Specifically, male participants, older individuals, married respondents, and those in the medicine field demonstrated higher knowledge scores compared to other groups (*p*-value < 0.001).

**Table 3 Participants' willingness to educate the public regarding Mpox (N = 437).**

| Respondent's intention | Strongly agree/Agree N (%) | Neutral N (%) | Strongly disagree/Disagree N (%) |
|---|---|---|---|
| Intention: I intend to educate the public regarding Mpox | 248 (56.8%) | 88 (20.1%) | 101 (23.1%) |
| Past experience: I educated the public regarding COVID-19 during pandemic | 273 (62.5%) | 75 (17.2%) | 89 (20.3%) |
| Subjective norm: People in my community want me to educate them about Mpox | 219 (50.1%) | 116 (26.5%) | 102 (23.3%) |
| Perceived self-efficacy: Whether or not to educate and counsel the public about Mpox is completely up to me | 223 (51.0%) | 94 (21.5%) | 120 (27.5%) |
| Perceived behavioral control: It is my responsibility to educate the community or public about Mpox | 262 (60.0%) | 82 (18.8%) | 93 (21.3%) |
| Attitude: I believe that the provision of education to the public about Mpox will help in disease prevention and control measures | 300 (68.6%) | 59 (13.5%) | 78 (17.8%) |

The participant's overall intention score was 21.39 ± 6.84 (range six to 30). The proportion of participants with high intention to educate the public regarding Mpox was 47.6% (scored > 23), 31.4% with moderate intention (scored 16–23), and 21.1% with low intention (scored < 16). The socio-demographics groups who showed more positive attitude than others were female, married, and those in the physiotherapy field ($p$-value ≤ 0.004). Table 4 summarizes the mean knowledge and intention scores among socio-demographics.

## Determinants of participants' knowledge and intention using ordinal logistic regression

The first analysis involved testing the assumptions of the OLR. The assumption of parallel lines, *i.e.*, the relationship between independent variables and dependent variables does not change according to the categories of the dependent variable, was achieved for both the knowledge and intention models. The $p$-values for the parallelism hypothesis testing, using the Chi-square test, were 0.999 and 0.065 for knowledge and intention models, respectively (>0.05). Each model's goodness of fit was tested using Chi-square (Pearson and Deviance) statistics. For the knowledge model, the $p$-values of 0.627 (Pearson Chi-square) and 0.995 (Deviance Chi-square) suggest that the model fits the data well and there is no discrepancy between the observed and expected values. Concerning the model fitting information, the $p$-values of the final model are 0.0002 and 0.005 (less than 0.05) for the knowledge and intention models, respectively. This indicates that the predictors significantly improve the model's fit compared to the model with no predictors.

For the knowledge model, there are six variables (age, gender, marital status, current role, educational level, and field of education) (Table 5). In the OLR analysis, the reference category was set as the last category for each independent variable. Examining the effect of independent variables on the knowledge as an outcome shows that only three categories of age and a master's degree (among the educational levels) have an impact on knowledge. For age, the estimates are negative (−1.984 for 20–25 years, −2.096 for 26–30 years, and −1.444 for 31–40 years) which indicates that participants in the lower age category are

**Table 4 Mean knowledge and intention scores among socio-demographics.**

| Variables | Knowledge of Mpox infection Mean (std. deviation) | Intention to educate the public regarding Mpox Mean (std. deviation) |
|---|---|---|
| Overall | 10.4 (4.88) | 21.39 (6.84) |
| Gender | | |
| Male | 11.34(4.84) | 20.39 (7.10) |
| Female | 9.51(4.76) | 22.32 (6.47) |
| *p*-value | **<0.001** | **0.003** |
| Age in years | | |
| 20–25 Years | 9.49(4.86) | 20.75(7.35) |
| 26–30 Years | 9.37(4.68) | 21.65(6.51) |
| 31–40 Years | 11.28(4.81) | 22.05(5.89) |
| 41–50 Years | 12.40(4.30) | 21.67(7.50) |
| >50 | 15.33(3.48) | 22.87(5.36) |
| *p*-value | **<0.001** | 0.4 |
| *Post hoc* analysis | 20–25 Years versus older age groups (more than 30) ≤**0.01** 26–30 Years versus older age groups (more than 40) ≤ **0.004** 31–40 Years versus older age group (more than 50) =**0.016** | |
| Marital status | | |
| Single | 9.76(4.90) | 20.18(7.18) |
| Married | 11.70(4.60) | 22.49(6.22) |
| Divorced | 7.71(4.22) | 21.67(6.20) |
| Widowed | 6.40(3.75) | 21.60(7.96) |
| *p*-value | **<0.001** | **=0.002** |
| *p*-value with *Post Hoc* tests | Married *versus* all others <**0.001** | Married *versus* Single = **0.004** |
| Current role | | |
| Student | 9.44(4.80) | 20.92(7.17) |
| Healthcare professional | 11.33(4.78) | 21.85(6.48) |
| *p*-value | **<0.001** | 0.2 |
| Education level | | |
| Student | 9.44(4.80) | 21.60(7.42) |
| Bachelor | 10.64(4.58) | 21.16(6.53) |
| Master | 10.69(5.07) | 21.26(6.42) |
| Doctorate (PhD) | 13.55(4.56) | 22.10(6.97) |
| *p*-value | **<0.001** | 0.8 |
| *p*-value with *Post Hoc* tests | Doctorate (PhD) versus all others ≤**0.035** | |
| Field of education | | |
| Pharmacy | 10.44(4.80) | 19.56(7.55) |
| Medicine | 12.06(4.56) | 21.79(6.28) |

*(continued on next page)*

**Table 4** (*continued*)

| Variables | Knowledge of Mpox infection Mean (std. deviation) | Intention to educate the public regarding Mpox Mean (std. deviation) |
|---|---|---|
| Dentistry | 8.13(4.41) | 19.83(8.09) |
| Nursing | 9.18(4.64) | 22.25(6.25) |
| Physiotherapy | 7.88(5.27) | 25.69(4.73) |
| Medical Laboratory Science | 9.38(5.28) | 23.66(6.56) |
| *p*-value | **<0.001** | **<0.001** |
| *p*-value with *Post Hoc* tests | Medicine *versus* Dentistry, Nursing and Physiotherapy = $\leq$ **0.01** | Nursing, Physiotherapy, Medical Laboratory Science versus Pharmacy $\leq$ **0.04** |

Notes.

Bolded values represent *p*-values less than 0.05.

more likely to have a lower level of knowledge compared to those aged more than 50 years old. In other words, the odds of having knowledge about Mpox increase with increasing age. For gender the estimate is positive (0.427 for male) which shows that being a male is associated with an increased likelihood of a participant falling into a higher category of knowledge compared to the female gender. The master's degree, as an educational level, has a negative estimate for the regression coefficient ($-1.037$, *p*-value $= 0.032$). For the impact of this category, participants with a master's degree tend to have a lower level of knowledge about Mpox as compared to those with a PhD degree. Being in the field of pharmacy (positive estimate $= 0.727$) or Medicine (positive estimate $= 0.709$) appears to positively impact the level of knowledge, however, the effect is insignificant as the *p*-values are 0.10 and 0.12 for pharmacy and medicine respectively.

For the intention model, the results of the OLR are given in Table 6. All the considered variables (total knowledge score, age, gender, marital status, current role, educational level, and field of education) in the OLR are found to be insignificant (all *p*-values are greater than 0.05). The total knowledge score has a very small positive estimate (0.001) which is insignificant (*p*-value $= 0.976$). All age categories have a negative estimate in predicting the intention when compared to the reference category, however, the effect is insignificant.

## DISCUSSION

An appropriate knowledge of healthcare professionals and students is of paramount importance to educate the community that will further decrease the transmission of Mpox infection. The viral transmission and the clinical presentation of Mpox is somewhat similar to other pox viruses such as smallpox and chickenpox (*Chen et al., 2005*; *Di Giulio & Eckburg, 2004*), therefore, the healthcare professionals must be updated and knowledgeable enough about Mpox to ensure that they can provide quality care for the local community. A previous study in Saudi Arabia detected an inadequacy in the knowledge and attitudes of the physicians regarding Mpox (*Alshahrani et al., 2022a*), similar to that of medical students (*Alshahrani et al., 2022c*), and Arabic regions (*Swed et al., 2023*), however, we did not come across any report assessing the knowledge of other healthcare professionals in

**Table 5  Parameter estimates of OLR using knowledge about Mpox as a response variable with three ordered categories ($n = 437$).**

|  |  | Estimate (Regression coefficient) | P-value | OR | 95% Confidence intervals of OR |
|---|---|---|---|---|---|
| Threshold | Poor knowledge | −0.513 | 0.669 | – | – |
|  | Average knowledge | 3.043 | 0.011 | – | – |
|  | Excellent knowledge (reference) | – | – | – | – |
| Independent variables | **Age in years** |  |  |  |  |
|  | 20–25 Years | −1.984 | **0.005** | 0.138 | 0.034–0.553 |
|  | 26–30 Years | −2.096 | **0.002** | 0.123 | 0.033–0.493 |
|  | 31–40 Years | −1.444 | **0.021** | 0.236 | 0.069–0.801 |
|  | 41–50 Years | −1.049 | 0.111 | 0.350 | 0.096–1.273 |
|  | *More than 50 (reference)* | – | – | – | – |
|  | **Gender** |  |  |  |  |
|  | Male | 0.427 | **0.041** | 1.533 | 1.130–2.351 |
|  | *Female (reference)* | – | – | – | – |
|  | **Marital Status** |  |  |  |  |
|  | Single | 1.117 | 0.221 | 3.056 | 0.510–18.301 |
|  | Married | 1.392 | 0.123 | 4.023 | 0.687–23.571 |
|  | Divorced | 0.487 | 0.63 | 1.627 | 0.224–11.834 |
|  | Widowed *(reference)* | – | – | – | – |
|  | **Current role** |  |  |  |  |
|  | Student | −0.08 | 0.771 | 0.923 | −0.616–0.457 |
|  | Healthcare professional (reference) | – | – | – | – |
|  | **Education level** |  |  |  |  |
|  | Student | −0.682 | 0.171 | 0.506 | 0.191–1.341 |
|  | Bachelor | −0.683 | 0.115 | 0.505 | 0.216–1.180 |
|  | Master | −1.037 | **0.032** | 0.355 | 0.137–0.917 |
|  | *Doctorate (PhD)–(reference)* | – | – | – | – |
|  | **Field of education** |  |  |  |  |
|  | Pharmacy | 0.727 | 0.101 | 2.069 | 0.867–4.938 |
|  | Medicine | 0.709 | 0.120 | 2.032 | 0.831–4.972 |
|  | Dentistry | −0.397 | 0.517 | 0.672 | 0.202–2.236 |
|  | Nursing | 0.462 | 0.324 | 1.587 | 0.634–3.970 |
|  | Physiotherapy | −0.511 | 0.492 | 0.600 | 0.140–2.577 |
|  | *Medical Laboratory Science (reference)* | – | – | – | – |

**Notes.**
Key: OR, Odds ratio.
Bolded values represent $p$-values less than 0.05.

the study area, therefore, this study included all healthcare professionals and students since they can play an important role in disease control.

This study appraised the knowledge as well as the intention of the healthcare professionals and students regarding the Mpox. The knowledge part included five domains: the general awareness about Mpox, the infection and viral transmission, the signs and symptoms, the periods of the disease and the prevention and treatment of the disease. The majority

**Table 6 Parameter estimates of OLR using the intention to educate the public about Mpox as a response variable with three ordered categories (n = 437).**

| | | Estimate (Regression coefficient) | P-value | OR | 95% Confidence intervals of OR |
|---|---|---|---|---|---|
| Threshold | Low intention | −2.473 | 0.016 | – | – |
| | Moderate intention | −0.959 | 0.347 | – | – |
| | High intention (reference) | – | – | – | – |
| | **Total knowledge score** | 0.001 | 0.976 | 1.001 | 0.961–1.042 |
| Independent variables | **Age in years** | | | | |
| | 20–25 Years | −0.727 | 0.287 | 0.483 | 0.127–1.842 |
| | 26–30 Years | −0.834 | 0.198 | 0.434 | 0.122–1.548 |
| | 31–40 Years | −0.721 | 0.236 | 0.486 | 0.148–1.600 |
| | 41–50 Years | −0.678 | 0.288 | 0.508 | 0.145–1.774 |
| | *More than 50 (reference)* | – | – | – | – |
| | **Gender** | | | | |
| | Male | −0.231 | 0.253 | 0.794 | 0.534–1.179 |
| | *Female (reference)* | | | | |
| | **Marital status** | | | | |
| | Single | −0.221 | 0.732 | 0.802 | 0.226–2.846 |
| | Married | 0.498 | 0.435 | 1.645 | 0.471–5.737 |
| | Divorced | 0.855 | 0.248 | 2.351 | 0.552–10.024 |
| | Widowed *(reference)* | – | – | – | – |
| | **Current role** | | | | |
| | Student | −0.265 | 0.286 | 0.767 | 0.472–1.249 |
| | Healthcare professional (reference) | – | – | – | – |
| | **Education level** | | | | |
| | Student | 0.533 | 0.245 | 1.704 | 0.694–4.187 |
| | Bachelor | 0.106 | 0.791 | 1.112 | 0.509–2.430 |
| | Master | −0.158 | 0.723 | 0.854 | 0.357–2.042 |
| | *Doctorate (PhD)–(reference)* | – | – | – | – |
| | **Field of education** | | | | |
| | Pharmacy | −0.767 | 0.062 | 0.464 | 0.207–1.041 |
| | Medicine | −0.459 | 0.281 | 0.632 | 0.274–1.456 |
| | Dentistry | −0.757 | 0.147 | 0.469 | 0.169–1.305 |
| | Nursing | −0.358 | 0.408 | 0.699 | 0.2991–1.632 |
| | Physiotherapy | 0.695 | 0.311 | 2.004 | 0.523–7.683 |
| | *Medical Laboratory Science (reference)* | – | – | – | – |

**Notes.**
Key: OR, Odds ratio.

of participants (77.1%) aware that Mpox is a public health emergency of international concern, however, less than half participants (45.1%) aware that the Mpox has been reported in Saudi Arabia with a considerable improvement from a previous report in which 95% of surveyed physicians stated that human Mpox case not reported in Saudi Arabia (*Alshahrani et al., 2022a*). This insufficiency in the general awareness of participants

regarding the Mpox in Saudi Arabia might be due to the fact that there were only a few cases reported at the time of conducting this survey (*Alshahrani et al., 2022b*).

The overall mean knowledge score was $10.4 \pm 4.88$ (range 0 to 21), and less than half participants were found to be with excellent or average knowledge. The overall significance of the model was evaluated using OLR. The model fitting information revealed a *p*-value of 0.0002 for the final knowledge model, indicating that the inclusion of predictors significantly improves the model's fit compared to a null model without predictors. By comparing the knowledge levels between health professionals and students using the ANOVA, notable variations in scores were observed across different educational backgrounds (*p*-value < 0.001), with healthcare professionals scoring higher than students. This difference may be attributed to healthcare professionals' exposure to actual cases and the ongoing pressure to update their knowledge due to clinical responsibilities. Similar trends have been observed in other studies from the Gulf region. For example, a study from the United Arab Emirates reported that the majority of health sciences students had poor knowledge about Mpox, underscoring a gap in preparedness at the educational level (*Abdelaziz Rashad Dabou, Magdi Ibrahim & Ekama Ilesanmi, 2024*). Furthermore, local studies in Saudi Arabia have highlighted knowledge gaps even among healthcare professionals. Previous research found that physicians' knowledge and attitudes towards Mpox were insufficient and varied by personal and professional factors, indicating areas for improvement (*Alshahrani et al., 2022a*). Besides, community pharmacists were found to possessed moderate knowledge of Mpox, particularly regarding clinical management, prevention, and vaccination, indicating room for improvement (*Alrasheedy et al., 2023*). These findings underscore the urgent need for targeted educational interventions across health professions in the Gulf region to ensure that all healthcare providers, including students, have access to up-to-date, evidence-based knowledge on Mpox diagnosis, prevention, and management. These findings on knowledge disparities are consistent with broader demographic trends. In our study, the male gender, older participants, married, healthcare professionals and those in medicine field were more knowledgeable than other groups (*p*-value < 0.001), more or less similar to the Nepalese healthcare workers (*Das et al., 2023*). This study demonstrates consistency between the ANOVA and OLR, indicating that age and gender significantly influence knowledge levels. Specifically, increasing age and being male are associated with a higher likelihood of attaining a higher level of knowledge. While older age, that is expected to impact clinical experience and practice, was a factor associated with participants' knowledge, consistent with previous studies conducted among Czech healthcare workers and Ghanaians, our findings contrast with these studies and with that conducted in Italy where females were found to be more knowledgeable in those contexts (*Boakye, Konadu & Mavhandu-Mudzusi, 2023*; *Miraglia Del Giudice et al., 2023*; *Riad et al., 2022*). Although more than seven in 10 participants recognized that Mpox is a viral infection, their correct answers across all questions regarding the infection and viral transmission ranged from average to poor. However, this is still much better than a previous study from 17 Arabic countries that reported less than one in 10 respondents correctly answered questions about the transmission route of Mpox (*Swed et al., 2023*). The failure to identify the mode of viral transmission might consequently lead to misdiagnosis

and treatment, and therefore, slow down the measures required to control the outbreak of Mpox. Likely, the participant's level of knowledge regarding the signs and symptoms of Mpox ranged between average to poor. There was also a poor level of knowledge regarding the incubation period, signs and symptoms period, whether past exposure to chickenpox can provide protection against Mpox, and the availability of specific drug or vaccine for Mpox. Even the clear sign of Mpox "rash" which expected to be recognized by all participants, was identified by only 64.8% as a symptom. Such level of knowledge was detected locally and internationally (*Alshahrani et al., 2022a*; *Harapan et al., 2020a*; *Sallam et al., 2022*; *Swed et al., 2023*). The reason behind this knowledge gap seems to be similar to other areas around the globe since some poxvirus diseases were eradicated and no longer included in the undergraduate curricula, in addition to the influence of the media on healthcare professionals' beliefs and knowledge with regard to the many reports and news published during the outbreak (*Harapan et al., 2020a*; *Riccò et al., 2022*). However, the fact behind the correct answer of more than two-thirds of participants about recommended prevention steps to protect people from Mpox might rely on common sense. Lacking essential disease information might confuse Mpox with related viral and non-viral diseases, and lead to inappropriate management, therefore, continues professional development programs and educational interventions seems to be necessary.

The overall intention score was $21.39 \pm 6.84$ (range 6 to 30), and the majority of participants were found to be with high or moderate intention to educate the public regarding Mpox. In a similar manner to the knowledge model, overall significance of the model was evaluated using OLR and the model fitting information revealed a *p*-value of 0.005 for the final intention model, indicating that the inclusion of predictors significantly improves the model's fit compared to a null model without predictors. Nonetheless, based on the OLR analysis, all the considered variables, including the total knowledge score, were found to be insignificant, with *p*-values greater than 0.05. In contrast, when considering single-variable models based on the ANOVA analysis, the female, married individuals, and those in the physiotherapy field showed more positive attitude than other groups (*p*-value $\leq$ 0.004). Analysis by education level revealed that individuals with a PhD demonstrated the highest intention to educate the public about Mpox (mean = 22.10, std. deviation = 6.97), however, the variation in intention scores among these educational groups was statistically insignificant (*p*-value = 0.8). Likely, healthcare professionals showed slightly higher intentions to educate the public about Mpox compared to students, with an average score of $21.85 \pm 6.48$ *versus* $20.92 \pm 7.17$ (*p*-value = 0.2), though the difference was not substantial. This pattern aligns with findings from a previous study, which showed that healthcare providers' knowledge, attitudes, and practices scores were significantly influenced by education level, among other factors (*Teng et al., 2022*). The skills, knowledge, and confidence of healthcare workers in diagnosing and managing Mpox cases could be influenced by the experience they gain from working in healthcare facilities with a high prevalence of the disease (*Harapan et al., 2020b*), and therefore, impact their attitude and intention to educate the public regarding Mpox. The analysis found that most participants had a past experience of educating the public for COVID-19 during pandemic and consequently, more than half of them are willing to continue educating the public

regarding this Mpox. Furthermore, the attitude, perceived subjective norm, self-efficacy and behavioral control were impacted around half or more participants, somewhat similar to the ratio of those with intention to educate the public about Mpox. This finding aligns with Ajzen's theory, which posits that the intention to perform a behavior, such as educating the public about Mpox, can be predicted by attitudes toward the behavior, perceived social norms, and the control that people perceive they have over their actions (*Hagger & Hamilton, 2024*; *Kan & Fabrigar, 2017*).

The collected data might be influenced by some limitations: more than half of the participants were not aware with Mpox cases in the study area and therefore, recall bias could affect the questions related to knowledge. The data included the ideas of only those using social media, selection bias, and therefore, findings from this survey cannot be generalized. This study did not follow up with the healthcare workers over time and hence, it is not possible to address the causal relationship of participants' factors with study outcomes. Furthermore, the survey did not evaluate the outcome of the participants' intentions. There is a need for case–control and cohort studies that consider the outcome of the knowledge on disease management and the outcome of the participants' intention to educate the public. Despite these limitations, the study is the first report in Saudi Arabia addressing the intention of healthcare workers to educate the public regarding Mpox.

## CONCLUSION

This study highlighted that most surveyed healthcare students and professionals acknowledged Mpox as a public health emergency of international concern and recognized the recommended prevention measures. However, knowledge levels regarding Mpox varied from average to inadequate, revealing deficiencies in specific information concerning transmission, symptoms, protection, and management. Notwithstanding these knowledge deficiencies, the majority of participants articulated moderate to high intentions to inform the public about Mpox, with many referencing prior experiences from the COVID-19 pandemic. The findings indicate that specialized educational programs centered on Mpox knowledge could improve preparedness among healthcare professionals. Additionally, offering supportive resources may enable healthcare students and professionals to engage more effectively in public health education and contribute to infection prevention initiatives.

## ACKNOWLEDGEMENTS

During the preparation of this work, the authors employed ChatGPT and Gemini to assist with language correction, clarity, and conciseness. Following the use of these tools, the authors conducted a thorough review and edit of the content, and take full responsibility for the content of the publication.

### Funding
This work was funded by the Deanship of Graduate Studies and Scientific Research at Jouf University under grant No. (DGSSR-2023-01-02340). The funders had no role in study design, data collection and analysis, decision to publish, or preparation of the manuscript.

### Grant Disclosures
The following grant information was disclosed by the authors:
The Deanship of Graduate Studies and Scientific Research at Jouf University: DGSSR-2023-01-02340.

### Competing Interests
The authors declare there are no competing interests.

### Author Contributions
- Malik Suliman Mohamed conceived and designed the experiments, performed the experiments, analyzed the data, authored or reviewed drafts of the article, and approved the final draft.
- Mona Timan Idriss analyzed the data, prepared figures and/or tables, and approved the final draft.
- Nasser Hadal Alotaibi analyzed the data, prepared figures and/or tables, and approved the final draft.
- Yusra Habib Khan analyzed the data, prepared figures and/or tables, and approved the final draft.
- Tauqeer Hussain Mallhi conceived and designed the experiments, analyzed the data, authored or reviewed drafts of the article, and approved the final draft.

### Human Ethics
The following information was supplied relating to ethical approvals (i.e., approving body and any reference numbers):
The Local Committee of Bioethics (LCBE) of Jouf University reviewed and granted an ethical approval with the number: 3-04-44.

### Data Availability
The raw data is available in the Supplemental File.

### Supplemental Information
Supplemental information for this article can be found online at http://dx.doi.org/10.7717/peerj.19162#supplemental-information.

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
