# Peer review of "Appraisal of healthcare students and professionals' knowledge and intention to educate the public regarding monkeypox (Mpox)"

_PeerJ, doi:10.7717/peerj.19162_

## Round 0.1 · original submission · Major Revisions

In addition to the comments of the reviewers, I found some gaps that should be addressed:

The manuscript should be updated, the term "mpox" is currently used in place of "monkeypox" .

The references should also be updated with the most recent studies.

The introduction should provide the hypothesis of the work by providing the necessity of the study. Mpox in Saudi Arabia? Studies conducted in SA ? ...

Reviewer 1 ·

Basic reporting

--The abstract notes "diverse communities" but are healthcare students and professionals "diverse?" There could be more connection between the study population, study aims, and the background.
--Educating the public about mpox or educating their patients about mpox?
--The authors missed in the introduction the mpox pandemic that sets the stage for the reason to do the study
--The authors made claims that are unsupported. For example, "It looks that the healthcare students and professionals might need to exert an extra effort in teaching and educating the public about the confirmed and potential source of transmission." but nothing before or after confirms this.
--What's the difference between knowledge and awareness?
--In addition, knowledge and awareness regarding mpox surrounding what? Transmission? vaccine?

Experimental design

--The abstract needs more information about what actual statistical methods were used (e.g., t-test, multiple logistic regression)
--Missing information on how participants heard about the study and were recruited into the study.
--The sample were healthcare professionals and students, so I am uncertain how 14.1% of the sample are unemployed professionals. Shouldn't they be removed from the analytical sample?
--More information is needed on how a composite knowledge and intention score were created to do the analyses in table 4.
--The authors used the Theory of Planned Behavior (see Table 3), however, there is no justification why they are using this theoretical framework.

Validity of the findings

--Conclusions not sufficiently supported by the data.
--Overreaching interpretations beyond the data.

Reviewer 2 ·

Basic reporting

Overall, the content of this manuscript is of good quality and high importance to publish to a wider audience. However, I notice so many typology and grammatical errors throughout the manuscript which hamper a fluent readability of this manuscript. I would suggest that the authors to re-read and revise all the errors by hiring a professional translator or grammar checker.

Experimental design

The experimental design of this manuscript is ok.

Validity of the findings

The validity of the findings is also well stated.

Additional comments

A review of the manuscript entitled “Appraisal of Healthcare students and professionals’ Knowledge and intention to educate the public regarding Mpox”

1. (page 6, line 27): Authors need to confirm that all acronyms are defined before being used for the first time. For example, WHO and CDC.
2. (page 6, lines 32-33): The percentage symbol (%) should be used unifiedly throughout the manuscript. The current usage is still inconsistent since sometimes the authors put a space in between the value and the percentage symbol. Please revise them accordingly.
3. (page 7, introduction part): Your introduction needs more information on the origin of monkeypox. I suggest to add more description about the discovery of monkeypox in 1958, when smallpox-like vesiculopustular lesions were observed on imported Java macaques that were kept in captivity. This information can be read from an article entitled “Monkeypox: Immune response, vaccination and preventive efforts”.
4. (page 7, line 62): “… hence the was named monkeypox” This sentence needs to be revised as it has a grammatical error in the word ordering.
5. (page 7, lines 81-82): “The illness ranges from less severe to more serious symptoms that need may require hospitalization.”
6. (page 8, lines 125-127): I noticed so many errors in this sentence, “The scientific information regarding Mpox was drived from the general information on the websites of the World Helath Orgnaziation (WHO) and Centere for Disease Control and Prevention (CDC)”. Please revise it. The word “drived” should be “derived”, “World Helath Orgnaziation” should be “World Health Organization”, and “Centere” should be “Centre”.
7. (page 8, lines 128-130): Instead of writing the sentences in their current version, it would be more effective to write them this way “Section I had socio-demographic information (6 items); Section II had general awareness questions about Mpox (2 items), the knowledge section (23 items) and the attitude section (6 items).”
8. (page 8, line 134): The word “and” is repetitive in this sentence.
9. (page 9, line 146): Please give the meaning of P, Q, and e in the equation.
10. (page 9, line 161): In my view, it would be better if the authors could explain the SPSS acronym earlier when they mention the statistical package of social sciences under the section of sample size estimation and data collection (line 151).
11. (page 10, line 182): the “ten” should be written as “10” here to adhere with the “7”. Remember to be consistent in your writing.
12. (page 10, line 184): what is virla infection? Please justify it since it suddenly appeared without prior explanation.
13. (page 10, line 192): there’s a typo in the word “awarness”.
14. (page 11, line 241): the word “hafs” should be revised to “has”.
15. (page 12, line 264): “gab” should be “gap”.
16. (page 12, lines 276-279): after mentioning the knowledge, skills, and attitude of the healthcare workers towards Mpox, it would be more interesting if the authors also mention about how general individuals with at least a BSc degree behave towards Mpox. In a study entitled “Global prevalence and determinants associated with the acceptance of monkeypox vaccination”, it’s stated that individuals with at least a BSc degree showed good knowledge and health behaviors.
17. (page 21, table 3): the “participants’ willing” should be changed to “participants’ willingness”.

Reviewer 3 ·

Basic reporting

The manuscript is clear and professional English.
The manuscript includes literature references however the last paragraph requires citations to support the claims made.

Experimental design

The manuscript is matched with the aim and scope of the journal.
The author needs to write the research question.
In the methods, the sampling size needs to be better justified with references, so the author should include a reference for the sample size calculation.
The author needs to explain the rationale behind selecting 509 participants out of the 545 total. This is slightly confusing to readers, and further clarification on the reasons for this decision would help eliminate any ambiguity.

Validity of the findings

In Table 2, the symbols "√" and "×" are used, but their meanings are not explained. The author should clarify these symbols under the table for the readers' understanding.
The findings presented in Table 4 are not discussed in the manuscript text. The author should provide a clear comment about it and discuss its findings in the discussion section to enhance the overall coherence of the manuscript.
The author should explicitly highlight the difference in knowledge levels between health professionals and students in the results section, and it should be emphasized clearly to present the data accurately since the title includes the students and professionals’ knowledge and intention.

Additional comments

It would also be beneficial if the author could include a discussion on the knowledge levels of students and the health profession in the Gulf region or between Arabs. There is abundant literature available from countries such as the UAE and Iraq, which can provide valuable context and comparisons.
https://doi.org/10.1177/23779608241256209
https://doi.org/10.1016/j.ijans.2024.100743
https://doi.org/10.3390/vaccines11030610

---

## Round 0.2 · Major Revisions

The authors provided considerable effort to improve the quality of the manuscript. However, some concerns raised, especially by reviewer 1, should be taken into consideration.

Reviewer 1 ·

Basic reporting

1. The literature review of healthcare professional knowledge about mpox is inadequate.
2. No justification for the constructs/variables
3. No background on the Theory of Planned Behavior, why this theory was used, nor any validity/reliability checks in measures. Knowledge is not a construct in the Theory of Planned Behavior, but is a background factor.
4. The discussion is more of a repeat of the results and mpox epidemiology than how key findings are concordant or discordant with the literature, implications for healthcare professional education/training, and future research directions.

Experimental design

1. I find the analytical plan to too simple and descriptive. Why not conduct a regression of how these factors contribute to healthcare professionals implementing an mpox practice?
2. The authors note that unemployed healthcare professionals ands students have an ethical obligation to protect the health of the public, but this is the authors' claims rather than an actual fact. I still want to see unemployed removed from the analysis. The authors did not provide needed justification for including unemployed professionals
3. If students do not have direct patient care, then they should also be removed from the analytical plan. The analytical plan and variables have little justification and many assumptions that are not grounded.

Validity of the findings

See notes above

Reviewer 2 ·

Basic reporting

no comment

Experimental design

no comment

Validity of the findings

no comments

Additional comments

I would like to compliment the authors for addressing most of my concerns. Now the manuscript has improved a lot and is simple to read through. However, I noted that in my previous review, the authors did not fully address my concerns. Therefore, I would like to highlight my suggestion here, which includes these points below:

1. (page 7, lines 70-71): This sentence “The Mpox virus was initially isolated in 1958 from Java macaques displaying symptoms of a rash and fever in captivity” will be more significant if it’s also supported by another relevant study. Please check 10.52225/narra.v2i3.90

2. (page 8, lines 123-125): This sentence also shares the same idea with a study entitled “Coping strategies used by healthcare professionals during COVID-19 pandemic in Dubai: A descriptive cross-sectional study” Kindly include this reference to make the sentence stronger.

Reviewer 3 ·

Basic reporting

the manuscript clear and containing sufficient background and literature references .

Experimental design

The authors need to write the research questions.

Validity of the findings

the statistical analysis is clear
conclusion well stated

---

## Round 0.3 · accepted · Accept

The authors have responded to all comments of the reviewers

Reviewer 2 ·

Basic reporting

no comment

Experimental design

no comment

Validity of the findings

no comment

Additional comments

no comment

Reviewer 3 ·

Basic reporting

literature references are sufficient

Experimental design

all required corrections were added

Validity of the findings

complete

Additional comments

all required corrections were done